# Psychometric Evaluation of the Nurses Professional Values Scale-3: Indonesian Version

**DOI:** 10.3390/ijerph18168810

**Published:** 2021-08-20

**Authors:** Asiandi Asiandi, Meli Erlina, Yu-Hua Lin, Mei-Chih Huang

**Affiliations:** 1International Doctoral Program in Nursing, Department of Nursing, College of Medicine, National Cheng Kung University, Tainan City 701, Taiwan; asiandi@ump.ac.id; 2Medical-Surgical Nursing Department, School of Nursing, Health Science Faculty, Universitas Muhammadiyah Purwokerto, Purwokerto 53181, Indonesia; melierlina1@gmail.com; 3Nursing Department, I-Shou University, Kaohsiung 824, Taiwan; lin2368@isu.edu.tw; 4Department of Nursing, College of Medicine, National Cheng Kung University, Tainan City 701, Taiwan; 5National Tainan Junior College of Nursing, Tainan City 700, Taiwan

**Keywords:** nurses, professional ethics, nursing ethics, reliability and validity, psychometrics, validation study

## Abstract

The Nurses Professional Values Scale-3 (NPVS-3) is an important instrument for measuring the development and sustainability of professional values in nurses and nursing students. The translation, adaptation, and validation on this scale, including in Indonesia, is still very limited. The purpose of this study was to examine the Indonesian version of NPVS-3. This study used forward and backward translation methods and a cluster random sampling of 600 nurses participated. The 28-item scale of NPVS3-I was tested using EFA, applying the principal axis factoring extraction method and varimax with Kaiser normalization rotation method. The CFA used SEM with AMOS. Findings suggested sufficient content validity, construct validity, and reliability of the Indonesian version of NPVS-3. The I-CVI values ranged between 0.80 to 1.00 and the S-CVI was 0.99. Construct validity was supported with loading factors ranging from 0.49 to 0.84 for three factors (Caring, Activism, and Professionalism). The CFA goodness-of-fit indices were *X*^2^ (*df*) = 1516.95 (347), *p* < *0*.001, normed chi-square (*X*^2^/*df*) = 4.37, RMSEA = 0.106, SRMR = 0.079, and CFI = 0.735. The Indonesian version of the NPVS3 showed good internal consistency with Cronbach’s alphas for the instrument of 0.97 and 0.94, 0.95, and 0.89 for Caring, Activism, and Professionalism, respectively. The Indonesian version of the NPVS-3 is valid and reliable for use in Indonesia.

## 1. Introduction

Professional value development is critical in a time of ever-increasing ethical dilemmas in care delivery [1]. Nurses are increasingly making professional judgments that can cause dilemmas, some of which lead to difficult ethical concerns [2] and ethical problems [3]. Recognizing values is essential for effectively responding to these ethical issues [2], particularly through the recognition of professional nursing values.

Professional values are conduct norms that clinicians and a professional need and that, therefore, provide context for judging performance [2]. Professional values and individual values have become the foundation of nurses’ conduct with patients [4]. The Nurses Professional Value Scale (NPVS) was the first scale used to measure the professional values of nurses developed by Weis and Schank [2]. This scale was later developed into the Nurses Professional Values Scale-Revised (NPVS-R) [1] and was finally updated to the Nurses Professional Values Scale-3 (NPVS-3) [5].

The NPVS was first developed based on the 1985 American Nurses Association Code of Ethics for Nurses with Interpretive Statements [2]. It is a scale with nine factors (caregiving, activism, accountability, integrity, trust, freedom, safety, and knowledge). This scale was later developed into the NPVS-R [1]. The NVPS-R was derived from the 2001 ANA Code of Ethics [1]. The NPVS-R has been culturally adapted and psychometrically tested in several countries. In Indonesia, this scale has been translated [6] and has shown excellent validity. Studies in Taiwan have also given similar results, as this scale was shown to be valid and reliable [7]. The NPVS-R has also been translated into Spanish, but the psychometric indicators have not yet been published [8]. In Italy, this scale showed greater internal consistency [9]. Weis and Schank admitted that the NPVS-R was not robust, so it is necessary to conduct further study on existing models [1]; then, they updated the NPVS-R to the NPVS-3, adjusting to changes of the ANA Code of Ethics for Nurses with Interpretive Statement [5].

NPVS-3 is a tool that can be used to assess the growth and advancement of professional values in nursing school and a variety of professional nursing contexts [5]. Efforts to understand the professional values of nursing are essential for the growth and development of personal freedom, integrity, compassion, and fairness [10]. Following Poorchangizi et al.’s [11] study, nurses also need to understand the most essential professional values in nursing related to preserving patient confidentiality, protecting the privacy rights of patients, being responsible for serving the needs of a multicultural society, and taking responsibility and accountability toward these activities. It was proved that an assessment of the growth and development of nursing professional values is very important for nurses, as evidenced by Poorchangizi et al. [12]; despite their greater expertise, nurses did not have markedly increased professional value mean scores compared to nursing students. Comparing the values of professional sophomores and senior bachelor nursing students, Posluszny and Hawley [13] found less evidence of linkage among the development of professional values and experience. These previous studies have proven the need for an assessment of the professional values of nurses using instruments that have been tested for validity and reliability, and follow the culture and characteristics of a particular nation. Due to the uniqueness and cultural differences between nations, we need a tool that can properly assess the professional values of nursing in each country, including in Indonesia. NPVS-3 is the only assessment tool that can be used after going through the process of translation and cultural adaptation. NPVS-3 has been adapted and tested for validity and reliability in Saudi Arabia and Italy [14,15] and has not yet been adapted into Bahasa Indonesia. Therefore, this study evaluated the psychometric properties of the Indonesian version of NPVS-3.

## 2. Materials and Methods

### 2.1. Participants

This study surveyed 600 nurses from a hospital in Central Java, Indonesia using NPVS-3. The sample was recruited based on the cluster random sampling technique. We randomized all the available wards using a random number generator and then selected nurses from the wards who were selected as participants in this study.

### 2.2. Instrument

The NPVS-3 used in this study, which was developed by Weis and Schank [5], is a scale that contributes to the measurement of nurses’ professional values. This scale is an updated version of the NPVS and NPVS-R [1,2], developed based on the 2015 ANA Code of Ethics for Nurses, which is a revision of the previous code of ethics. The latest code of ethics focuses on the fundamental values and commitments of nurses, as well as with the additional roles of nurses which have expanded to include roles in the population and responsibilities for patient health and safety. The NPVS-3 is an instrument with 28 items using a Likert scale format between 1 (*not important*) to 5 (*most important*). Each item is a descriptive phrase that describes a specific code provision and its interpretive commentary, all in the form of positive phrases. The possible scores obtained are between 28 to 140, with a higher score representing a stronger values orientation of the nurse. The total score is the result of the sum of all items. The original NPVS-3 research was conducted on 1139 participants consisting of baccalaureate nursing students (*n* = 243), graduate nursing students (*n* = 237), and nurses (*n* = 659). The results of the study have shown an internal consistency of three factors (Caring, Activism, and Professionalism) between 0.80 to 0.91 and the overall scale alpha coefficient was 0.90. Using the minimum factor loading criteria of 0.30 for each retain item [16], it was found that three factors explained 51.62% of the extracted common variance. Confirmatory factor analysis (CFA) using goodness-of-fit indices shows the root mean square error of approximation (RMSEA) is 0.065, comparative fit index (CFI) is 0.90, and goodness-of-fit index (GFI) is 0.875, showing the adequacy of fit [5].

### 2.3. Procedures

The translation of the Nurse Professional Values Scale-3 (NPVS-3) was carried out in several stages following guidelines of the translation process, adaptation, and validation from Sousa and Rojjanasrirat [17]. In the first stage, the original/source language version of NPVS-3 was translated by two bilingual translators who are fluent in the source language and the target language according to the specifications. One translator came from the Language Development Center (LDC) at the Muhammadiyah University of Purwokerto (UMP) and another translator was a nursing expert, a nursing lecturer at the Muhammadiyah University of Purwokerto. From this forward translation, two versions of the Indonesian NPVS-3 were found.

The second stage was the comparison stage of translation results by translator I (TL-1) and translator II (TL-2). The translation results were compared to find out which translation was better to be used as the Indonesian version of the NPVS-3, synthesis I version. The third stage was back translation (B-TL) or translation back from the NPVS-3 target language to the source language. The back translation stage was also carried out by two translators. One translator came from the UMP LDC and was a different person than the forward translation stage, and another translator was a nursing expert, a nursing lecturer at Kusuma Husada University Surakarta. Interpreters did not know the results of each translation. The fourth stage was the comparison of B-TL I and B-TL 2. From this back-translation process, two versions of back translation were produced, then it was determined which translation was better to be used as synthesis II.

The fifth stage was the monolingual trial. This trial stage was carried out on nurses other than the hospital nurses who carried out the full psychometric test. The trial was carried out on 20 nurses who were willing to fill out the online questionnaire with clear and unclear choices, then at the end of the questionnaire, there was a column provided for participants to provide input, criticism, and suggestions. This process was done to determine the level of understanding and clarity of the Indonesian version of the NPVS-3, to improve the ambiguity of words on the scale. Then, to determine equality, a content validity test was carried out by 10 experts who were selected according to the criteria as an expert panel.

### 2.4. Data Collection

The questionnaire was given to nurses in the ward through the head nurse (HN) of each ward. The questionnaire was sent sealed using an envelope and accompanied by a cover letter addressed to the participants. Participants then filled out the questionnaire and returned it in a sealed envelope through the HN of each ward within two weeks. Returning the questionnaire and filling out the questionnaire completely were considered as a willingness to participate in this study.

### 2.5. Data Analysis

We utilized IBM SPSS for Windows version 25.0 (IBM Corp., Armonk, NY, USA) and IBM AMOS for Windows version 24.0 (IBM Corp., Armonk, NY, USA). Descriptive analysis was applied for the demographic data. The *Kolmogorov-Smirnov* test of the data distribution analysis was showed unmeet the normal distribution of data. We applied the *Mann-Whitney U* test for the variable with two categorical and *Kruskal-Wallis K* test for the variable with more than two categorical. The content validity was assessed by 10 expert panel members [18]. The construct validity was assessed for convergent and discriminant validity [19,20]. The convergent validity was assessed by computing the average variance extracted (AVE) for every construct by summed square correlation (*R*^2^) for each item and by dividing it by the total number of items. An AVE higher than 0.50 supports the convergent validity of the construct [19,21]. To assess discriminant validity, we calculate the shared variance between constructs from the correlation between constructs then squared this correlation index, and the values should be less than the AVE [20]. Furthermore, we applied the Heterotrait–Monotrait Ratio of Correlation (HTMT) as a procedure to double-check the discriminant validity between constructs, using a cut-off less than 0.85 that denotes fixed discriminant validity and indicates the difference between each construct [22]. We calculated the HTMT using a plugin for AMOS created by James Gaskin [23]. The internal consistency was assessed as reliability for each subscale and all items by using Cronbach’s alpha and the construct/composite reliability (CR). The content validity was assessed for the item content validity index (I-CVI) and the scale content validity index (S-CVI) by using a 4-point rating scale, where 1 = *irrelevant* and 4 = *extremely relevant* [18,24].

Construct validity was evaluated by exploratory factor analysis (EFA) and confirmatory factor analysis (CFA). For this analysis, we split data into two independent data sets [25,26,27]. In the first stage, EFA was performed for the principal analysis factor with varimax rotation on the first half of the independent sample. The first half of the independent sample was used in EFA as an initial analysis for the CFA model [25]. The preliminary analysis results assessed for the correlation matrix and scanned the matrix for correlations greater than 0.30, to check for the correlation matrix greater than this value. We should be aware of a value greater than 0.90 for any multicollinearity problem [28]. The testing assumptions of EFA were determined with the Kaiser–Meyer–Olkin (KMO) measure of sampling adequacy and Bartlett’s test of sphericity. The KMO minimum criterion of 0.50 and Bartlett’s measure should be statistically significant (*p* < 0.50) [20,28]. Kaiser’s criterion of retaining factors used eigenvalues greater than 1. The factor loading significance considers a loading of 0.30 [20]. When variables were found to show significant loading in more than one factor, the problem of cross-loading exists. This would be identified using a ratio between the larger variance and the smaller variance. A ratio between 1.0 to 1.5 is problematic cross-loading, between 1.5 to 2.0 is potential cross-loading, and greater than 2.0 is ignorable cross-loading [20]. To assess the variables which meet the acceptable levels of explanation we used the criterion of communality (squared multiple correlations among variables) 0.50 or greater [20,29].

In the second stage, the EFA results were confirmed for a rigorous empirical item set by using EFA for the second half of the independent sample [24]. Some indices were used in CFA as follow: The absolute fit indices chi-square (*X*^2^) test, the normed chi-square (*X*^2^/*df*), the root mean square error of approximation (RMSEA), the standardized root mean square residual (SRMR), and the comparative fit index (CFI) [20,22]. The chi-square fit index evaluates how well a hypothesized model fits data from a set of measuring items [30]. Multivariate normality of data, acceptable sample size, no structured incomplete data, and sufficient model specification are all requirements of the chi-square model fit index [30]. The indices threshold as recommended by Hu and Bentler [31] and Gaskin and Lim [32], is as follows: the normed chi-square >3 is acceptable and >1 is excellent; RMSEA >0.06 is acceptable and <0.06 is excellent; SRMR >0.08 is acceptable and <0.08 is excellent; and CFI <0.95 is acceptable and >0.95 is excellent. 

### 2.6. Ethical Considerations

This study was approved by the institutional review board of the health research ethics commission from the associated university (Approval No. KEPK/UMP/07/VI/2020) and obtained permission from the hospital where this study was conducted.

## 3. Results

### 3.1. Demographic Variables, Work-Related Variables, and Nurses’ Professional Values

A total of 600 nurses participated in this study and were separated into two data sets. The demographic characteristics of participants are summarized in Table 1. The participant means age was 33.24 ± 7.54 years. Most participants were female (63.30%) and the main religion was Muslim (99.50%). Most of the participants were married (67.20%). Over half of the participants’ education level was a diploma (56.80%) and more were associate nurses (59.60%) with full-time jobs (87.70%). Most of the participants had a working experience of between 5 to 14 years (57.20%).

The participant age had a significant negative correlation with Caring (*r_s_* = −0.218), Activism (*r_s_* = −0.184), and Professionalism (*r_s_* = −0.168), *p* < 0.001 for each variable, respectively. There was no significant difference in mean of Caring, Activism, and Professionalism for gender, *p* > 0.05 for each variable, respectively. Similarly, there was no significant difference in mean rank of Caring, Activism, and Professionalism for religion, *p* > 0.05 for each variable, respectively. There was a significant difference in mean of Caring and Activism for marital status, *H*(2), *p* = 0.025 and *H*(2) = 7.15, *p* = 0.028, respectively; however, there was no significant difference in mean of Professionalism, *H*(2) = 7.36, *p* = 0.135. There was a significant difference in mean for Caring, Activism, and Professionalism for nurse education level, *H*(2) = 9.71, *p* = 0.008; *H*(2) = 16,72, *p* < 0.001; and *H*(2) = 15.40, *p* < 0.001, respectively. There was a significant difference in mean of Caring, Activism, and Professionalism for nurse’s expertise, *H*(3) = 18.72, *p* < 0.001; *H*(3) = 24.79, *p* < 0.001; and *H*(3) = 17.24, *p* = 0.001, respectively. There was no significant difference in mean of Caring, Activism, and Professionalism for type of job, *U* = 18,403.50, *z* = −0.760, *p* = 0.447; *U* = 17102.50, *p* = 0.089; and *U* = 16683.00, *p* = 0.046, respectively. There was a significant difference in mean of Caring, Activism, and Professionalism for working experience, *H*(2) = 21.22, *p* < 0.001; *H*(2) = 10.56, *p* =0.005; and *H*(2) = 11.72, *p* = 0.003, respectively.

### 3.2. Validity Assessment

The validity test was carried out by testing the scale on 10 panels of experts in the field of nursing and the bilingual test involved the translation of the Indonesian version of the NPVS3-I with four choices of rating scales (1 = very irrelevant, 2 = somewhat relevant, 3 = relevant, 4 = very relevant) [15,21]. This test was conducted to assess the conceptual equality between items on the Item Content Validity Index (I-CVI) and a scale to assess the Scale Content Validity Index (S-CVI). The results of the content validity of the NPVS3-I items ranged from 0.80 to 1.0, and the S-CVI value was 0.99. This suggested a good content validity index [18,24,33].

The NPVS3-I validity was tested using principal axis factor analysis on 28 items with varimax rotation. The Kaiser–Meyer–Olkin measurement verified the adequacy of the sample to be analyzed, KMO = 0.96 (*p* < 0.001), and Bartlett’s test of sphericity was statistically significant (*p* < 0.001), indicating sample adequacy [28,33]. Initial analysis was carried out to obtain eigenvalues of each factor in the data. Three factors had more than 1 eigenvalue according to the Kaiser criteria and accounted for 65.40% of the extracted variance. The factor loading ranged from 0.49 to 0.84 and the communality ranged from 0.49 to 0.84 (see Table 2).

The AVEs for Caring, Activism, and Professionalism were 0.36, 0.48, and 0.33, respectively, which were not supported by the convergent validity for the indicators of each unobservable variable [19,20]. The shared variance between Caring and Activism was (0.922)^2^ = 0.85, Caring and Professionalism was (0.948)^2^ = 0.90, and Activism and Professionalism was (0.932)^2^ = 0.87, respectively. The shared variances were not supported by the discriminant validity, as it was greater than the AVE values. The HTMT value between the constructs of Caring and Activism, Caring and Professionalism and Activism and Professionalism were 0.90, 0.97, and 0.94, respectively. These three constructs were nearly indistinguishable and greater than the threshold of 0.85, indicating that they did not support discriminant validity [23].

The CFA confirmed the three factors solution for the model: Caring, Activism, and Professionalism (see Figure 1). The indices indicated an acceptable fit based on the following goodness-of-fit indices: *X*^2^ (*df*) = 1516.95 (347), *p* < 0.001, normed chi-square (*X*^2^/*df*) = 4.37, root mean square error of approximation (RMSEA) = 0.106, standardized root mean residual (SRMR) = 0.079, and comparative fit index (CFI) = 0.735.

### 3.3. Reliability Assessment

The internal consistency for the NPVS3-I was good. The Cronbach’s alpha for the instrument was 0.97. Cronbach’s alpha for each factor was 0.94, 0.95, and 0.89, respectively. Similarly, the construct reliability (CR) for each factor was 0.86, 0.90, and 0.77, supporting a good internal consistency. The corrected item-total correlation coefficient ranged from 0.56 to 0.86. The squared multiple correlations (SMCs) values ranged from 0.38 to 0.78. The coefficient alpha if the item was deleted ranged from 0.86 to 0.94 (see Table 3). The comparison of Cronbach’s alpha and items distribution in the original and new model can be seen in Table 4. 

## 4. Discussion

The psychometric properties of NPVS3-I were well established among nurses in Indonesia. The CVI was accepted based on the recommendation. The recommended value for the I-CVI should be no less than 0.78 for six to ten experts and a minimum of 0.90 for S-CVI [18,24,34].

The EFA in this study resulted in three factors, the same as the original scale. The number of items in this study was similar to the scale of the original items (28 items). This study obtained three factors, as follows: factor 1 (Caring), factor 2 (Activism), and factor 3 (Professionalism). In EFA, the explanatory constructs are referred to as factors (or latent variables), and typically reflect clustering of variables that are strongly correlated or have a greater factor loading [28]. The factor structure in this study was distributed into the three factors, similar to the original version [5] and the Arabic version (NPVS3-A) [14]. The factor loading ranged from 0.49 to 0.84, indicating 24% to 71% variance (fair to excellent variance) [29,35] with the communality ranging from 0.49 to 0.84, where greater communalities indicate a better correlation coefficient in a more reliable sample size [20,29,36]. The communality is squared factor loadings that indicate the item’s reliability [37].

Caring is the first factor in NPVS3-I, accounting for 57.91% of the variance in the nurses’ professional values. Caring is an integral part of the first three sections of the nursing code (the 2015 ANA Code of Ethics for Nurses) [38]. These sections are devoted to the patient as an individual, family, or population, personal health, patient health protection, and unbiased patient care [5]. Activism is the second factor of NPVS3-I, accounting for 4.42% of the variance in the nurses’ professional values. This factor is an essential part of the last three sections of the code [38], which highlight the nurse’s activist role, including portions of tasks that go far beyond individual patient interactions. This factor concentrates on basic freedoms, a worldwide recognition of the nature of humanity, which includes ecological and sustainability justice concerns and the practice’s role in creating public policy, professional contributions in supporting global health, helping to reduce health inequalities, engagement in nursing associations, and contributing to research and scholarly exploration [5]. The third factor, professionalism, accounted for 3.07% of the variance in the nurses’ professional values. This factor focuses on the code’s fourth through sixth sections [38], which deal with the limits of duty and loyalty. Autonomy, accountability, and responsibility for care delivery, leading in health promotion, the obligation for career development and well-being, and providing ethical and excellent treatment in a supportive environment are all covered in these parts [5].

The CFA indices indicated model acceptable fits for chi-square value and normed chi-square value, and an excellent fit for SRMR [31]. The indices of CFI and RMSEA were non-significant fit since both these indices related to the need for significantly larger sample size [20]. This model accounted for 65.40% of the common variance extracted in NPVS3-I. It supported the satisfactory construct validity [20]. A recent study showed a higher accounted variance than the original version with common variance extracted at 51.62% [5], which was lower than the Arabic version (NPVS3-A) with a common variance extracted of 67.50% [14].

Both the current study and the original version of the NPVS-3 have a similarity in the number of factors obtained (Caring, Activism, and Professionalism). However, the item distribution showed the displacement of items into other factors and it was different from the original version. In the current study, we moved Item 28, Item 9, and Item 8 from Factor 1 (Caring) to Factor 3 (Professionalism). We also moved Items 10, 11, 12, and 17 from Factor 1 (Caring) to Factor 2 (Activism), moved Item 7 and Item 1 from Factor 2 (Activism) to Factor 3 (Professionalism), and moved Items 2, Item 3, and Item 15 from Factor 3 (Professionalism) to Factor 1 (Caring), considering the nature of the original scale [5] and based on the results of discussions from the authors of the current study by considering the suitability of the content of each of these items.

Construct validity showed the AVE values for the constructs Caring, Activism, and Professionalism were less than 0.50, which did not support convergent validity for any unobservable variable [19,20]. The shared variance between Caring and Activism, Caring and Professionalism, and Activism and Professionalism did not support discriminant validity. The HTMT value did not support the discriminant validity between constructs of Caring and Activism, between Caring and Professionalism, and between Activism and Professionalism, where the results were greater than the threshold of 0.85 [20]. The three constructs shared variance, indicating non-differentiable constructs [23,31,39].

The current study showed a higher internal consistency both for the scale and the sub-scales (see Table 3 and Table 4). This corresponds with the original version in internal consistency with Cronbach’s alpha 0.94 [5] and the Arabic version with Cronbach’s alpha 0.97 [14]. Internal consistency for each of the sub-scales Caring, Activism, and Professionalism for the current study was 0.94, 0.95, and 0.89, respectively; the original version was 0.89, 0.91, and 0.80, respectively; and the Arabic version was 0.97, 0.96, and 0.89, respectively. The recommendation acceptable value for internal consistency was 0.70 and above for Cronbach’s alpha [18,25,34]. The corrected item-total correlation coefficient ranged from 0.59 to 0.84. The squared multiple correlations (SMCs) values ranged from 0.43 to 0.76; the larger values of SMCs indicate more stable factors [29].

Several limitations should be considered in this study. First, the use of a self-report questionnaire might cause limitations because of the halo effect or response set biasing [40,41]. Second, the AVE for Caring, Activism, and Professionalism was lower than cutoff 0.50 [19,20], which was not supported by the convergent validity for the indicators of each unobservable variable. Third, Caring and Activism, Caring and Professionalism, and Activism and Professionalism were non-differentiable constructs that may cause limitations due to the lack of discriminant validity for these three constructs. Third, non-robust models in the current study indicate the need for further studies to find a better model.

## 5. Conclusions

The results of the current study provide evidence that the NPVS3-I is valid and reliable, so it can be used as an instrument to measure nurses’ professional values in Indonesia. The NPVS3-I has three factors similar to the original version (Caring, Activism, and Professionalism), with any differences in item distribution between the original version and the Indonesian version possibly due to the existence of socially and culturally differences in participants. We recommend the use of NPVS3-I to measure nurses’ and nursing students’ professional values in Indonesia. This instrument also can be used as a screening tool pre-and post-training related to nurses’ professional values.

## Figures and Tables

**Figure 1 ijerph-18-08810-f001:**
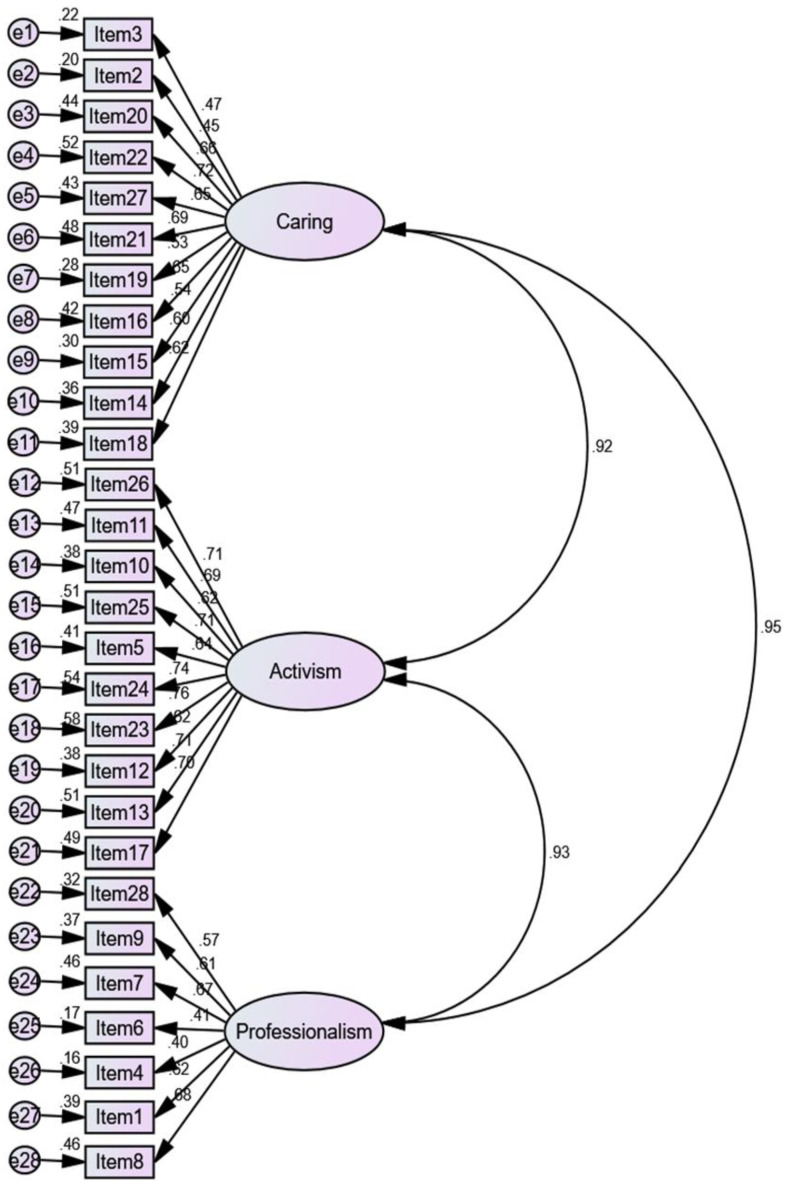
Confirmatory factor analysis for the Indonesian version of the Nurses Professional Values Scale-3 (*n* = 300).

**Table 1 ijerph-18-08810-t001:** Demographic variables, work-related variables, and nurses’ professional values (*n* = 600).

Variables	EFA	CFA	Total	Caring			Activism			Professionalism		
	N1 (%)	N2 (%)	N (%)	*M* (*SD*)	*r_s_/U/H*	*p*	*M* (*SD*)	*r_s_/U/H*	*p*	*M* (*SD*)	*r_s_/U/H*	*p*
Age (years), *M (SD)*	33.12 (7.87)	33.36 (7.22)	33.24 (7.54)	36.51 (4.95)	−0.218	<0.001	35.84 (5.54)	−0.184	<0.001	29.05 (4.08)	−0.168	<0.001
Gender					36743.50	0.434		36684.00	0.415		37144.00	0.563
Male	90 (30.00)	94 (31.30)	184 (30.70)	36.25 (5.44)			35.54 (5.59)			28.92 (4.04)		
Female	210 (70.00)	206 (68.70)	416 (63.30)	36.62 (4.72)			35.98 (5.53)			29.11 (4.09)		
Religion					830.50	0.828		834.00	0.836		853.00	0.887
Muslim	299 (99.70)	298 (99.30)	597 (99.50)	36.50 (4.95)			35.83 (5.52)			29.05 (4.06)		
Christian	1 (0.30)	2 (0.70)	3 (0.50)	37.67 (6.66)			38.00 (10.44)			30.00 (7.94)		
Marital status					7.36	0.025		7.15	0.028		4.01	0.135
Married	234 (78.00)	169 (56.30)	403 (67.20)	36.19 (4.95)			35.55 (5.44)			28.92 (3.98)		
Not married	65 (21.70)	129 (43.00)	194 (32.30)	37.07 (4.88)			36.34 (5.69)			29.28 (4.26)		
Divorce	1 (0.30)	2 (0.70)	3 (0.50)	42.67 (4.04)			43.33 (3.51)			33.00 (2.00)		
Nurse education level					9.71	0.008		16.72	<0.001		15.40	<0.001
Diploma	171 (57.00)	170 (56.70)	341 (56.80)	35.85 (4.87)			34.97 (5.25)			28.52 (4.06)		
Bachelor’s	129 (43.00)	128 (42.70)	257 (42.80)	37.34 (4.90)			36.96 (5.68)			29.73 (3.97)		
Master’s	0 (.00)	2 (.70)	2 (0.30)	41.00 (9.90)			42.00 (11.31)			34.00 (7.07)		
Nurse’s expertise					18.72	<0.001		24.79	<0.001		17.27	0.001
Advanced nurse	90 (30.00)	76 (25.30)	166 (27.70)	35.53 (5.12)			35.11 (5.47)			28.78 (4.05)		
Supervisor nurse	26 (8.70)	22 (7.30)	48 (8.00)	35.60 (4.26)			34.88 (4.95)			28.56 (4.33)		
Associate nurse	155 (51.70)	202 (67.30)	357 (59.50)	36.89 (4.93)			35.92 (5.49)			29.01 (3.96)		
Senior nursing student	29 (9.70)	0 (0.00)	29 (4.80)	38.97 (3.96)			40.66 (5.25)			32.03 (4.25)		
Type of job					18403.50	0.447		17102.50	0.089		16683.00	0.046
Full time	258 (86.00)	268 (89.30)	526 (87.70)	36.46 (4.82)			35.69 (5.41)			28.91 (3.94)		
Part-time	42 (14.00)	32 (10.70)	74 (12.30)	36.86 (5.83)			36.97 (6.33)			30.11 (4.82)		
Working experience (years)					21.22	<0.001		10.56	0.005		11.72	0.003
<5	54 (18.00)	85 (28.30)	139 (23.20)	38.12 (5.65)			36.75 (6.10)			29.74 (4.45)		
5–14	189 (63.00)	154 (51.30)	343 (57.20)	36.28 (4.70)			35.95 (5.52)			29.09 (4.04)		
≥15	57 (19.00)	61 (20.30)	118 (19.70)	35.29 (4.29)			34.47 94.62)			28.14 (3.53)		

Note. EFA, exploratory factor analysis; CFA, confirmatory factor analysis; *M* (*SD*), mean (standard deviation); *r_s_*, Spearman’s correlation coefficient; *U*, Mann–Whitney test; *H*, Kruskal–Wallis test.

**Table 2 ijerph-18-08810-t002:** Factor loading and communality of the Indonesian version of the Nurse Professional Values Scale-3 (NPVS3-I) (*n* = 300).

Items	Factor			Communality
	Caring	Activism	Professionalism	
Item 3	0.84			0.82
Item 2	0.82			0.81
Item 20	0.69			0.73
Item 22	0.69			0.74
Item 27	0.66			0.73
Item 21	0.68			0.65
Item 19	0.66			0.76
Item 16	0.63			0.58
Item 15	0.55			0.61
Item 14	0.54			0.67
Item 18	0.49			0.54
Item 26		0.76		0.72
Item 11		0.72		0.72
Item 10		0.70		0.69
Item 25		0.68		0.72
Item 5		0.67		0.74
Item 24		0.63		0.79
Item 23		0.63		0.66
Item 12		0.58		0.67
Item 13		0.59		0.72
Item 17		0.56		0.65
Item 28			0.75	0.69
Item 9			0.74	0.68
Item 7			0.68	0.75
Item 6			0.60	0.67
Item 4			0.59	0.59
Item 1			0.58	0.63
Item 8			0.53	0.59
Eigenvalue	16.21	1.24	0.86	
Percentage of variance	57.91%	4.42%	3.07%	
Total variance explained of the factor model			65.40%	
Kaiser–Meyer–Olkin measure of sampling adequacy			0.96	
Bartlett’s test of sphericity		*X* ^2^	7867.74	
		*df*	378	
		*p*	<0.001	

Note. *df*, degree of freedom; *p*, probability. Factor loadings less than 0.30 have not been printed and variables have been sorted by loadings.

**Table 3 ijerph-18-08810-t003:** Item means, standard deviation, corrected item to total correlations, squared multiple correlations, and alpha if item deleted for the NPVS3-I (*n* = 300).

Items	Mean	*SD*	CITC	SMCs	AID
Factor 1: Caring (Cronbach’s *α* = 0.94)					
Item 3. Protect health and safety of the patient/public	4.04	0.70	0.67	0.67	0.93
Item 2. Respect the inherent dignity, values, and human rights of individuals	4.02	0.71	0.68	0.68	0.93
Item 20. Confront practitioners with questionable or inappropriate practice	3.70	0.73	0.77	0.73	0.93
Item 22. Practice guided by principles of fidelity and respect for person	3.82	0.74	0.81	0.72	0.92
Item 27. Engage in consultation/collaboration to provide optimal care	3.74	0.76	0.76	0.65	0.93
Item 21. Protect rights of participants in research	3.81	0.74	0.76	0.66	0.93
Item 19. Safeguard patient’s right to confidentiality and privacy	4.12	0.76	0.75	0.67	0.93
Item 16. Act as a patient advocate	3.53	0.71	0.64	0.55	0.93
Item 14. Accept responsibility and accountability for own practice	3.66	0.72	0.71	0.60	0.93
Item 15. Protect moral and legal rights of patients	4.01	0.76	0.79	0.73	0.93
Item 18. Provide care without bias or prejudice to patients and populations	3.69	0.73	0.71	0.52	0.93
Factor 2: Activism (Cronbach’s α = 0.95)					
Item 26. Take action to influence legislators and other policy makers to improve health care	3.56	0.78	0.75	0.62	0.94
Item 11. Recognize the role of professional nursing associations in shaping health policy	3.78	0.75	0.78	0.69	0.94
Item 10. Advance the profession through active involvement in health-related activities	3.80	0.69	0.74	0.63	0.94
Item 5. Participate in peer review	3.63	0.72	0.72	0.53	0.94
Item 25. Promote mutual peer support and collegial interactions to ensure quality care and professional satisfaction	3.60	0.70	0.81	0.69	0.94
Item 24. Participate in professional efforts to advance global health	3.76	0.75	0.86	0.78	0.94
Item 23. Actively promote health of populations	3.69	0.73	0.78	0.67	0.94
Item 12. Establish collaborative partnerships to reduce health care disparities	3.74	0.72	0.77	0.64	0.94
Item 13. Assume responsibility for meeting health needs of diverse populations	3.70	0.73	0.80	0.65	0.94
Item 17. Participate in nursing research and/or implement research findings appropriate to practice	3.59	0.71	0.75	0.60	0.94
Factor 3: Professionalism (Cronbach’s *α* = 0.89)					
Item 28. Recognize professional boundaries	3.77	0.75	0.63	0.46	0.88
Item 9. Seek additional education to update knowledge and skills to maintain	3.81	0.73	0.66	0.48	0.87
Item 6. Establish standards as a guide for practice	3.89	0.75	0.70	0.54	0.87
Item 7. Promote and maintain standards where planned learning activities for students take place	3.73	0.80	0.78	0.64	0.86
Item 1. Engage in ongoing self-evaluation	3.59	0.72	0.68	0.47	0.87
Item 4. Assume responsibility for personal well-being	3.71	0.79	0.56	0.38	0.88
Item 8. Initiate actions to improve environments of practice	3.64	0.81	0.74	0.59	0.86
Overall Cronbach’s *α* = 0.97					

Note. CITC, corrected item-total correlation; SMCs, squared multiple correlations; AID, alpha if item deleted.

**Table 4 ijerph-18-08810-t004:** Cronbach’s alpha and items distribution in the original and new model.

Factor	Original Model		Model after EFA	
	Items	Reliability	Items	Reliability
Caring	15, 18, 19, 2, 3, 22, 16, 14, 21, 20	0.885	3, 2, 20, 22, 27, 21, 19, 16, 15, 14, 18	0.937
Activism	24, 23, 26, 12, 13, 11, 10, 17, 25, 27	0.912	26, 11, 10, 25, 5, 24, 23, 12, 13, 17	0.946
Professionalism	6, 7, 5, 8, 1, 9, 4, 28	0.799	28, 9, 7, 6, 4, 1, 8	0.886
Overall		0.944		0.974

## Data Availability

The data presented in this study are available on request from the corresponding author. The data are not publicly available due to privacy and ethical reason.

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
