# Peer review of "Psychometric Evaluation of the Nurses Professional Values Scale-3: Indonesian Version"

_ijerph, 2021, doi:10.3390/ijerph18168810_

Round 1

Reviewer 1 Report

Congratulations on the research

The introduction is not well thought out. Much of the content of the introduction refers to information about the questionnaire, which should be covered in the methods section. There is insufficient substantiation of the indications for the use of the questionnaire, it is based mainly on the psychometric properties of the questionnaire and its versions.

The initial questionnaire explained in the text was validated in different groups of nurses (students, baccalaureate, professionals,...) Is there any difference in this respect in this current questionnaire that has been validated with a sample of 600 nursing professionals?

Line 75 is written in the future, and is not correct.

What is the possible explanation proposed by the authors for the three factors that finally make up the questionnaire after the factor analysis? Why these three factors? What significance do the authors give to this fact?

Reviewer 2 Report

Thank you for writing the paper as a good study. As a tool validation study on nursing professionalism, it is thought to be highly useful as basic data for research on nurses. 

  1. Introduction: Please be specific about why you chose this tool to measure nursing professionalism.
  2. Discussion: Most of the discussion is about result descriptions. It is necessary to compare and analyze the tool evaluation with other versions of the tool or the tool evaluating the nursing professionalism. Please elaborate further on this.

Reviewer 3 Report

Thank you for sending your paper entitled “Psychometric Evaluation of the Nurses Professional Values Scale-3: Indonesian Version” to IJERPH. After carefully review this interesting paper, the following comments are listed for your reference:

  1. Abstract: To increase potential citations, authors should check keywords against those recommended in the MeSH Browser of Medical Subject Headings https://meshb.nlm.nih.gov/. For example: “professional values; instrument; development; validity; reliability” are not MeSH. I recommend that you change this keywords.
  2. Introduction (lines 75-76): The objective must be written in past not future verb tense.
  3. Result: Table 1. Were statistically significant differences found between demographic variables? Specify it

Round 2

Reviewer 2 Report

Thank you for reviewing your research.